# Regional excess mortality during the 2020 COVID-19 pandemic in five European countries

Garyfallos Konstantinoudis [1✉], Michela Cameletti[2], Virgilio Gómez-Rubio [3], Inmaculada León Gómez[4,5], Monica Pirani [1], Gianluca Baio[6], Amparo Larrauri[4,5], Julien Riou[7], Matthias Egger[7,8], Paolo Vineis[1] & Marta Blangiardo[1]

The impact of the COVID-19 pandemic on excess mortality from all causes in 2020 varied across and within European countries. Using data for 2015–2019, we applied Bayesian spatio-temporal models to quantify the expected weekly deaths at the regional level had the pandemic not occurred in England, Greece, Italy, Spain, and Switzerland. With around 30%, Madrid, Castile-La Mancha, Castile-Leon (Spain) and Lombardia (Italy) were the regions with the highest excess mortality. In England, Greece and Switzerland, the regions most affected were Outer London and the West Midlands (England), Eastern, Western and Central Macedonia (Greece), and Ticino (Switzerland), with 15–20% excess mortality in 2020. Our study highlights the importance of the large transportation hubs for establishing community transmission in the first stages of the pandemic. Here, we show that acting promptly to limit transmission around these hubs is essential to prevent spread to other regions and countries.

[1] MRC Centre for Environment and Health, Department of Epidemiology and Biostatistics, School of Public Health, Imperial College London, London, UK. [2] Department of Economics, University of Bergamo, Bergamo, Italy. [3] Departamento de Matemáticas, Escuela Técnica Superior de Ingenieros Industriales, Universidad de Castilla-La Mancha, Albacete, Spain. [4] National Centre of Epidemiology (CNE), Institute of Health Carlos III, Madrid, Spain. [5] Consortium for Biomedical Research in Epidemiology and Public Health (CIBERESP), Institute of Health Carlos III, Madrid, Spain. [6] Department of Statistical Sciences, University College London, London, UK. [7] Institute of Social and Preventive Medicine, University of Bern, Bern, Switzerland. [8] Population Health Sciences, Bristol Medical School, University of Bristol, Bristol, UK. ✉email: g.konstantinoudis@imperial.ac.uk

By December 2020, the World Health Organization (WHO) reported 1,813,188 Coronavirus disease 2019 (COVID-19) related deaths globally[1]. Although COVID-19 related deaths are key to monitoring the pandemic's burden, vital statistics generally suffer from issues related to accuracy and completeness[2]. Additionally, they can be subject to changes in definition and different policies regarding testing and reporting[3,4]. At the same time, focusing only on COVID-19 deaths does not provide information about indirect pandemic effects due to disruption to health services and wider economic, social and behavioural changes in the population[5]. An effective way to quantify the total mortality burden of the COVID-19 pandemic is through excess mortality[6]. Excess mortality compares the number of deaths from all causes observed during the pandemic, and the number of deaths expected had the pandemic not occurred, using data from recent pre-pandemic years. Preliminary estimates suggest that the total number of global deaths attributable to the COVID-19 pandemic in 2020 is at least 3 million, with approximately 37% of these deaths occurring in the European region[1].

Previous studies have examined excess mortality at the national level reporting a disproportionate mortality burden[6,7]. In Europe, England and Wales, Spain and Italy experienced the largest increase in mortality during March-May 2020, with excess mortality estimates for the two sexes ranging from 70 to 102 per 100,000 population[7]. In contrast, from October to December 2020, the impact on mortality was great in Switzerland[6]. Studies focusing on the regional level have also found differential effects on mortality[8–10]. In Italy higher excess mortality was observed in some provinces in the North-west[8,9]. In England, the highest excess mortality was observed in London and in the West Midlands during March–May 2020[11,12]. In Greece, Eastern Macedonia and Thrace, Western Macedonia and Central Macedonia reported more than 10% excess mortality, in contrast to the Aegean islands and Crete for which the excess was smaller than 3%[10]. Studying these variations may help our understanding of the transmission patterns and the effectiveness of policies and measures to contain the pandemic. Other factors that may have also contributed to the varying impact on mortality across regions include differences in demographics[13], the prevalence of comorbidities[13] and environmental factors[12,14,15].

In this study of five European countries, we examined the impact of the COVID-19 pandemic on mortality in 2020 using weekly regional all-cause mortality data. We included countries from Northern, Western and Southern Europe (England, Greece, Italy, Spain and Switzerland) and analysed regions defined by Eurostat, which are consistent across different European countries, Supplementary Fig. 1. We used a model-based approach to predict deaths for 2020 by specific age- and sex groups, under the counterfactual scenario that COVID-19 had not occurred. To the best of our knowledge, this is the first international study of the COVID-19 regional impact on mortality.

## Results

A total of 565,505 deaths were recorded in 2020 in England, 132,514 in Greece, 756,450 in Italy, 485,536 in Spain and 77,222 in Switzerland (Table 1). The estimated population in 2020 was 56,702,967 in England, 10,718,447 in Greece, 59,641,219 in Italy, 47,332,587 in Spain and 8,681,297 in Switzerland (Table 1). In all countries, the number of deaths in males compared to females was larger for all age groups below 80 years. Comparing the observed number of deaths with the mean number of deaths from 2015 to 2019, there were 40,631 and 27,739 excess deaths in males and in females, respectively, in England, 5,380 and 5,909 in Greece, 59,327 and 52,206 in Italy, 35,868 and 33,208 in Spain and 5788 and 4526 in Switzerland (Table 1).

**Table 1 Age and sex-specific number of excess and observed deaths and population for 2020 by country of death. The expected number of deaths were calculated as the mean of crude deaths by age and sex during 2015-2019. The population numbers refer to the 1st of January in 2020.**

| | England | | | Greece | | | Italy | | | Spain | | | Switzerland | | |
|---|---|---|---|---|---|---|---|---|---|---|---|---|---|---|---|
| | Excess | Observed | Population | Excess | Observed | Population | Excess | Observed | Population | Excess | Observed | Population | Excess | Observed | Population |
| **Males** | | | | | | | | | | | | | | | |
| <40 | −1216 | 6675 | 14,400,227 | −139 | 1230 | 2,477,962 | −616 | 4652 | 12,067,152 | −114 | 4040 | 10,318,712 | 57 | 898 | 2,054,813 |
| 40–59 | 2850 | 27,448 | 7,285,268 | 90 | 5976 | 1,511,414 | 1819 | 26,463 | 9,063,297 | 479 | 22,786 | 7,436,645 | −50 | 2855 | 1,257,300 |
| 60–69 | 3407 | 36,749 | 2,860,281 | 773 | 9151 | 606,322 | 4749 | 42,224 | 3,527,676 | 3806 | 32,930 | 2,576,976 | 186 | 4385 | 470,892 |
| 70–79 | 12,039 | 72,721 | 2,323,869 | 855 | 15,023 | 455,459 | 14,173 | 90,103 | 2,736,665 | 10,216 | 57,285 | 1,810,637 | 1386 | 8900 | 347,460 |
| ≥80 | 23,551 | 142,090 | 1,182,213 | 3801 | 35,476 | 164,268 | 39,201 | 204,874 | 1,655,296 | 21,482 | 129,962 | 1,056,287 | 4210 | 21,061 | 178,639 |
| Total | 40,631 | 285,683 | 28,051,858 | 5380 | 66,856 | 5,215,425 | 59,327 | 368,316 | 29,050,086 | 35,868 | 247,003 | 23,199,257 | 5788 | 38,099 | 4,309,104 |
| **Females** | | | | | | | | | | | | | | | |
| <40 | −488 | 4142 | 13,913,794 | −86 | 538 | 2,434,019 | −367 | 2466 | 11,469,522 | −14 | 2265 | 9,957,901 | −4 | 479 | 1,965,192 |
| 40–59 | 1308 | 17,636 | 7,443,579 | 211 | 3075 | 1,603,581 | 512 | 15,611 | 9,288,376 | 357 | 11791 | 7,432,715 | −72 | 1676 | 1,242,592 |
| 60–69 | 1403 | 24,300 | 3,009,520 | 485 | 4502 | 685,346 | 1401 | 23,441 | 3,836,725 | 2180 | 15553 | 2,773,818 | 0 | 2508 | 485,305 |
| 70–79 | 7529 | 54,344 | 2,583,842 | 308 | 9899 | 548,506 | 5934 | 58,466 | 3,231,926 | 5223 | 32827 | 2,173,945 | 791 | 6125 | 395,491 |
| ≥80 | 17,987 | 179,400 | 1,700,375 | 4992 | 47,644 | 231,571 | 44,726 | 288,150 | 2,764,583 | 25,462 | 176,097 | 1,794,951 | 3811 | 28,335 | 283,613 |
| Total | 27,739 | 279,822 | 28,651,110 | 5909 | 65,658 | 5,503,023 | 52,206 | 388,134 | 30,591,133 | 33,208 | 238,533 | 24,133,330 | 4526 | 39,123 | 4,372,194 |
| **Total** | | | | | | | | | | | | | | | |
| <40 | −1705 | 10,817 | 28,314,021 | −226 | 1768 | 4,911,980 | −983 | 7718 | 23,536,674 | −128 | 6305 | 20,276,614 | 53 | 1377 | 4,020,006 |
| 40–59 | 4158 | 45,084 | 14,728,847 | 301 | 9051 | 3,114,996 | 2331 | 42,074 | 18,351,674 | 836 | 34,577 | 14,869,360 | −122 | 4531 | 2,499,892 |
| 60–69 | 4809 | 61,049 | 5,869,801 | 1258 | 13,653 | 1,291,668 | 6151 | 65,665 | 7,364,402 | 5985 | 48,483 | 5,350,794 | 186 | 6893 | 956,197 |
| 70–79 | 19,568 | 127,065 | 4,907,711 | 1163 | 24,922 | 1,003,965 | 20,107 | 148,569 | 5,968,591 | 15,439 | 90,112 | 3,984,582 | 2177 | 15,025 | 742,950 |
| ≥80 | 41,537 | 321,490 | 2,882,587 | 8793 | 83,120 | 395,839 | 83,927 | 493,024 | 4,419,878 | 46,944 | 306,059 | 2,851,238 | 8021 | 49,396 | 462,252 |
| Total | 68,368 | 565,505 | 56,702,967 | 11,289 | 132,514 | 10,718,447 | 111,532 | 756,450 | 59,641,219 | 69,077 | 485,536 | 47,332,587 | 10,314 | 77,222 | 8,681,297 |

**Model validation**. Overall, the models had good predictive ability in cross validation over 2015 to 2019, Supplementary Table 1. The highest correlation between observed and predicted was observed for age group 80 years or older, with the medians and 95% credible intervals (i.e. 0.95 probability that the true value lies in this interval) ranging from 0.83 (95% CrI 0.82–0.84) in females in England to 0.97 (95% CrI 0.97–0.98) in males in Spain. For the same age group, the coverage (i.e., the probability that the observed number of deaths falls in the 95% credibility interval of the predicted) varied from 0.90 in females in Spain to 0.95 in males in Switzerland, Supplementary Table 1. Although the coverage for the <40 age group is close to 0.95, we excluded this age group from the results reported, as (i) the correlation between observed and predicted values was low, varying from 0.15 in females England to 0.69 in males in Spain, Supplementary Table 1, and (ii) this was the age group least affected by COVID-19 mortality during 2020[13]. In line with the literature[16], the relationship between temperature and mortality was U-shaped, steeper for colder temperatures and stronger for the older people, Supplementary Figs. 2–6.

**Country-level trends and overall excess mortality**. Temporal patterns differed across countries, with England and Spain experiencing a larger death toll during March-May 2020 and Switzerland and Greece during November–December 2020, Supplementary Figs. 2–3 and 5–6. Mortality was high in Italy during both periods, Supplementary Fig. 4. Across the five nations, the median relative excess mortality together with the 95% credible intervals for the entire 2020 (relative to what is expected had the pandemic not occurred) ranged from 6% (95% CrI −1–13%) in Greece to 12% (95% CrI 6%–19%) in Spain in men. For women, it ranged from 6% (95% CrI −2%–13%) in Greece to 12% (95% CrI 6%–19%) in Spain, Fig. 1 and Supplementary Tables 2–6. For the country-level weekly trends see Supplementary Supplementary Figs. 7 to 11.

**Sub-national level patterns: NUTS2 regions**. Excess mortality was evident for most regions, but with large intra-country variability, as shown in Fig. 1. Across the five countries, Madrid, Castile-La Mancha, Castile-Leon (Spain) and Lombardia (Italy) are the regions with the highest excess mortality in 2020, ranging from 28% (95% CrI 22%–34%) to 33% (95% CrI 27%–39%) for males and 25% (95% CrI 17%–38%) to 32% (95% CrI 23%–40%) for females (Fig. 1 and Supplementary Tables 4–5). Ceuta (Spain) experienced a similar median excess for females (31%: 95% CrI 14%–54%), albeit associated with larger uncertainty, Fig. 1 and Supplementary Table 5. For males, the regions most affected in England, Greece and Switzerland were Outer London and the West Midlands, Eastern, Western and Central Macedonia, and Ticino, with the median excess mortality varying from 15% (95% CrI 8%–24%) to 21% (95% CrI 15%–28%), Fig. 1 and Supplementary Tables 2–3 and 6. For females, the median excess mortality varied from 18% for the Ticino to 19% in Western Macedonia and 15% in East-North East outer London, Fig. 1 and Supplementary Tables 2 and 6. The regions that show the lowest excess mortality are Cornwall and Isle of Scilly and Devon in England, Crete, and North and South Aegean in Greece for males, Lazio in Italy for females and Canary and the Balearic islands in Spain, Fig. 1 and Supplementary Tables 2–6. Overall, the estimates of the excess mortality in males and females were similar, with the excess being consistently lower for females in England, Supplementary Table 2 and Fig. 1.

**Finer sub-national level patterns: NUTS3 regions**. The higher resolution maps in Figs. 2 and 3 show the median relative excess mortality and the posterior probability of a positive excess, allowing appreciation of patterns missed by the lower resolution. In England, the high excess experienced by the West Midlands was driven by Birmingham, the largest urban area in the region, which recorded values above 20%. In Greece, the municipality of Drama, with an excess >20% was responsible for the high excess in Eastern Macedonia and Thrace. For Italy, outside the Northern regions, the model highlights localised excess in the provinces of Rimini, Pesaro-Urbino and Foggia, on the Eastern coast, while central Spain and Catalonia had the highest excess in Spain. In Switzerland, all regions had a relative excess of <20% but the French and Italian speaking regions experienced a posterior probability of a positive excess above 0.95. In contrast, the German-speaking regions were more diverse, Figs. 2 and 3. In Supplementary Figs. 12 to 21 we report the spatial trends by age and sex at the higher geographical resolution. The spatial trends differ depending on the age group, but are similar for men and women for age groups over 60 years.

**Spatio-temporal trends at the regional level**. Figure 4 shows the relative excess mortality and the posterior probability that the excess mortality is greater than 0 for the different NUTS2 regions, for each week in 2020. Across the five countries, relative excess larger than 200% is observed only during the first epidemic period of 2020 (March–May 2020) in England (Greater London), Italy (Lombardia) and Spain (Madrid, Castille-La Mancha, Catalonia), Fig. 4. In Switzerland, during the first epidemic period, the geographical patterns of excess mortality were highly localised, with Ticino experiencing the highest excess mortality. In contrast, the geographical variability in Greece was more diffuse. During the second epidemic period (October–December 2020), the excess mortality in Italy and Switzerland was similar across the country, whereas in Greece it was highly localised, with Central Macedonia experiencing a relative excess mortality between 100–200% during November 2020, Fig. 4. Supplementary Figs. 22–41 show the half-year median excess mortality by age, sex and NUTS3 regions and the posterior probability that the excess is larger than 0. Across the five countries the north of Italy and central Spain were hit the most during the first 6 months of 2020. Greece was mainly affected during the second 6 months of the pandemic with the North of the country and the older people most affected, Supplementary Figs. 26–29.

## Discussion

To the best of our knowledge, this is the first multi-country study examining excess mortality in 2020 across five European countries at the sub-national level. We found that excess mortality in 2020 varied widely both between countries and within countries. Spain experienced the largest excess mortality among the five countries studied. Within Greece and Italy the northern regions were more affected than other regions. The temporal trends at the sub-national level showed patterns of localised excess mortality in England, Italy, Spain and Switzerland during the first wave, whereas in Greece the excess mortality was homogeneous. During the second wave, excess deaths were overall lower in magnitude and their distribution more homogeneous in England, Italy and Spain. In contrast, in Greece and Switzerland, the second wave was more severe than the first one.

Our study has several strengths. It quantifies the short term, direct effect of the COVID-19 pandemic and indirect effects on mortality due to other life-threatening conditions, such as myocardial infarction[17], in five European countries. Reduced access to or uptake of medical care due to COVID-19 leading to delays in cancer screening, cancer diagnosis, rescheduling of surgery or cancellation of outpatient visits in patients with chronic

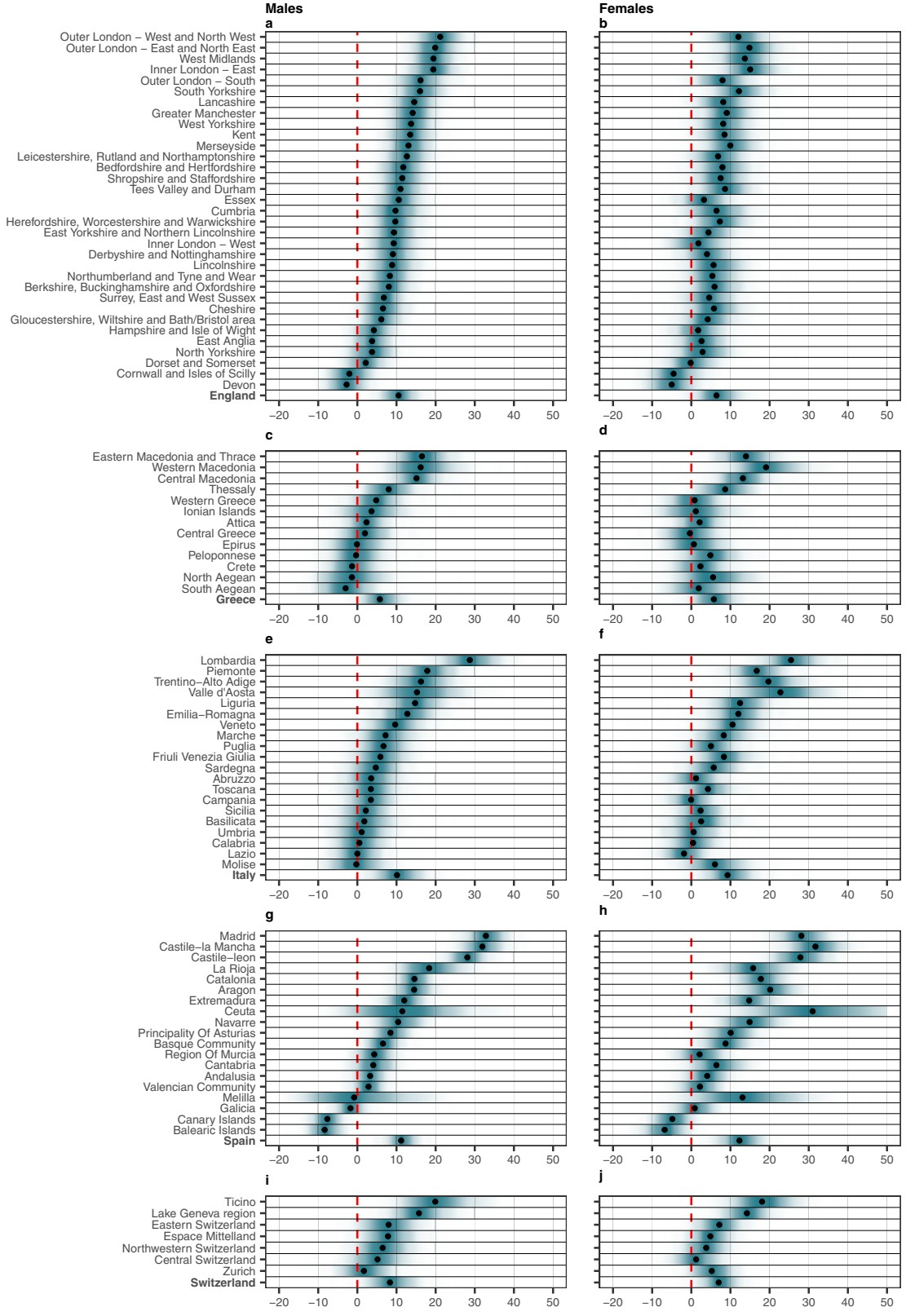

conditions may have increased mortality during the pandemic, particularly in countries with weaker health systems[18–20]. In contrast, a population-based study in the UK found that that primary care contacts for physical and mental health conditions decreased after the introduction of lockdowns in March, 2020,

and remained below pre-lockdown levels[21]. These trends, could represent a substantial burden of unmet needs, with potential implications for subsequent morbidity and premature mortality[21]. Our modelling approach accounts for spatial and temporal mortality trends, factors such as temperature and public

**Fig. 1 Posterior distribution of relative excess deaths (%) across the different countries by NUTS2 region and sex in 2020. a** Posterior distribution of relative excess deaths (%) in males in England. **b** Posterior distribution of relative excess deaths (%) in females in England. **c** Posterior distribution of relative excess deaths (%) in males in Greece. **d** Posterior distribution of relative excess deaths (%) in females in Greece. **e** Posterior distribution of relative excess deaths (%) in males in Italy. **f** Posterior distribution of relative excess deaths (%) in females in Italy. **g** Posterior distribution of relative excess deaths (%) in males in Spain. **h** Posterior distribution of relative excess deaths (%) in females in Spain. **i** Posterior distribution of relative excess deaths (%) in males in Switzerland and **j** posterior distribution of relative excess deaths (%) in females in Switzerland. The black dots represent the medians of the posterior distribution of relative excess deaths. The red line highlights the 0% relative excess deaths, which means no observed difference in the 2020 mortality compared to the counterfactual scenario that the pandemic did not occur.

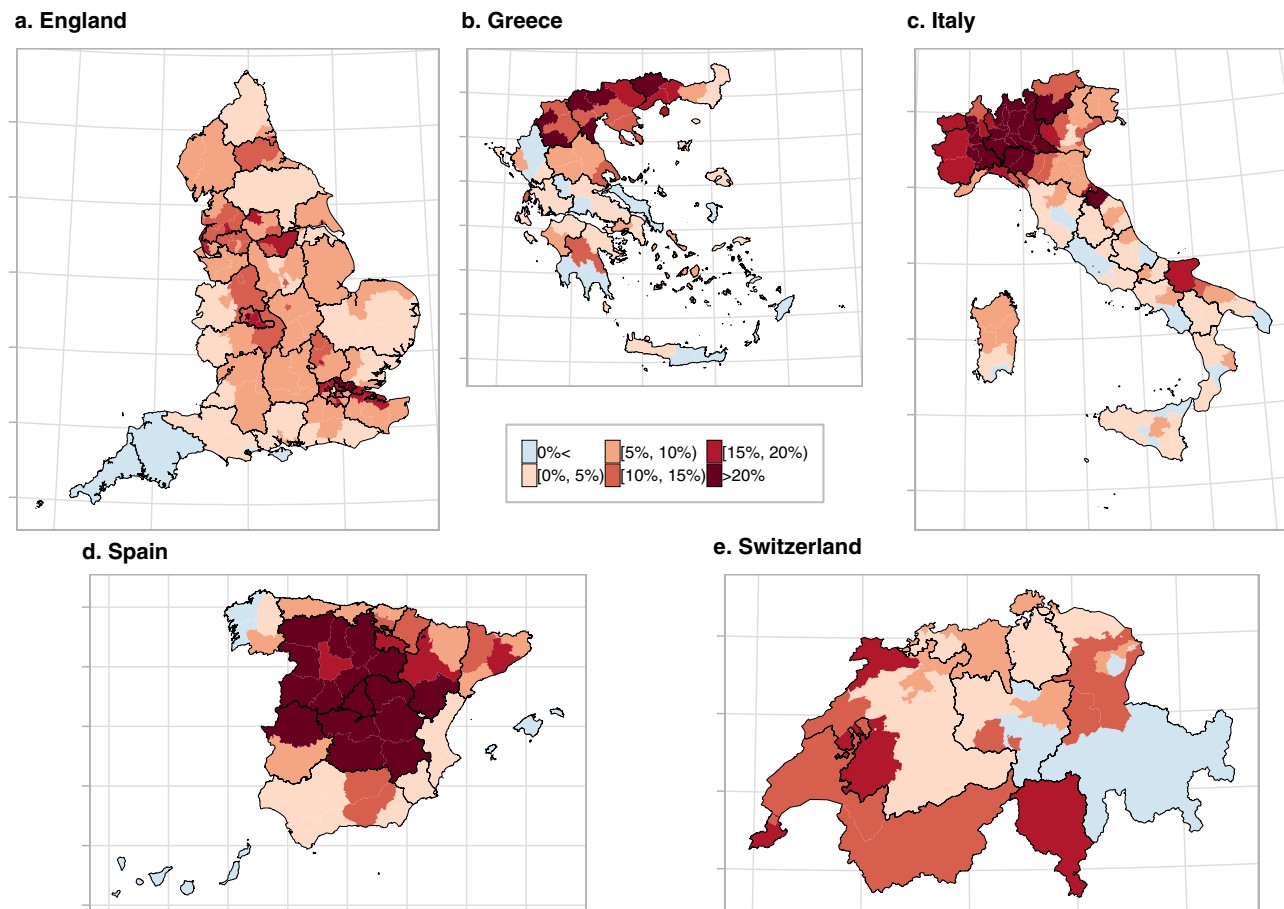

**Fig. 2 Median relative excess deaths (%) by NUTS3 region in 2020. a** Median relative excess deaths (%) in England. **b** Median relative excess deaths (%) in Greece. **c** Median relative excess deaths (%) in Italy. **d** Median relative excess deaths (%) in Spain and **e** median relative excess deaths (%) in Switzerland in categories. Areas in blue indicate areas that observed less deaths than expected had the pandemic not occurred, whereas the different shades of red indicate the higher relative excess mortality. The black solid lines correspond to the NUTS2 region borders.

holidays, and the different population temporal trends across space, age and sex groups, factors not taken into account in previous reports[10,11]. We carefully validated the model employing a cross-validation approach and found that it had high predictive accuracy. In contrast to previous studies, we stratified by age and sex, thus allowing the spatial and temporal mortality trends to vary across these groups. Weaknesses include the lack of detailed data on the causes of death, which would have allowed insights into the sources of the observed variation in excess deaths. We used population estimates coming from the national statistical institutes, but we did not account for their uncertainty, which is supposed to increase the further we move from the last census. This is likely to underestimate the width of the 95% CrI of the excess mortality[22].

Several previous studies reported nationwide excess mortality for 2020. The Office for National Statistics in England reported a 17.9% increase in male mortality and 11.2% in females[11]. A recent study of 40 industrialised countries covered the period from February 2020 to February 2021 and found an excess mortality of 15% to 20% in England and Wales, Spain and Italy[23]. Our point estimates are lower but when we consider their uncertainty, the differences are relatively small and probably explained by the periods used to train the model, the data sources used, the prediction periods and the different population estimation approaches[24]. Other data-related differences may play a role: for instance, previous analyses considered England and Wales[23], while our focus is on England only. Our results are in line with estimates from the Hellenic Statistical Authority, which reported a 7.3% increase in the relative excess in Greece during 2020[10], a Swiss study reporting a 10.6% increase in excess mortality in males and a 7.2% increases in females relative to 2019[25] and the estimates from the Italian National Institute of Statistics[26]. The

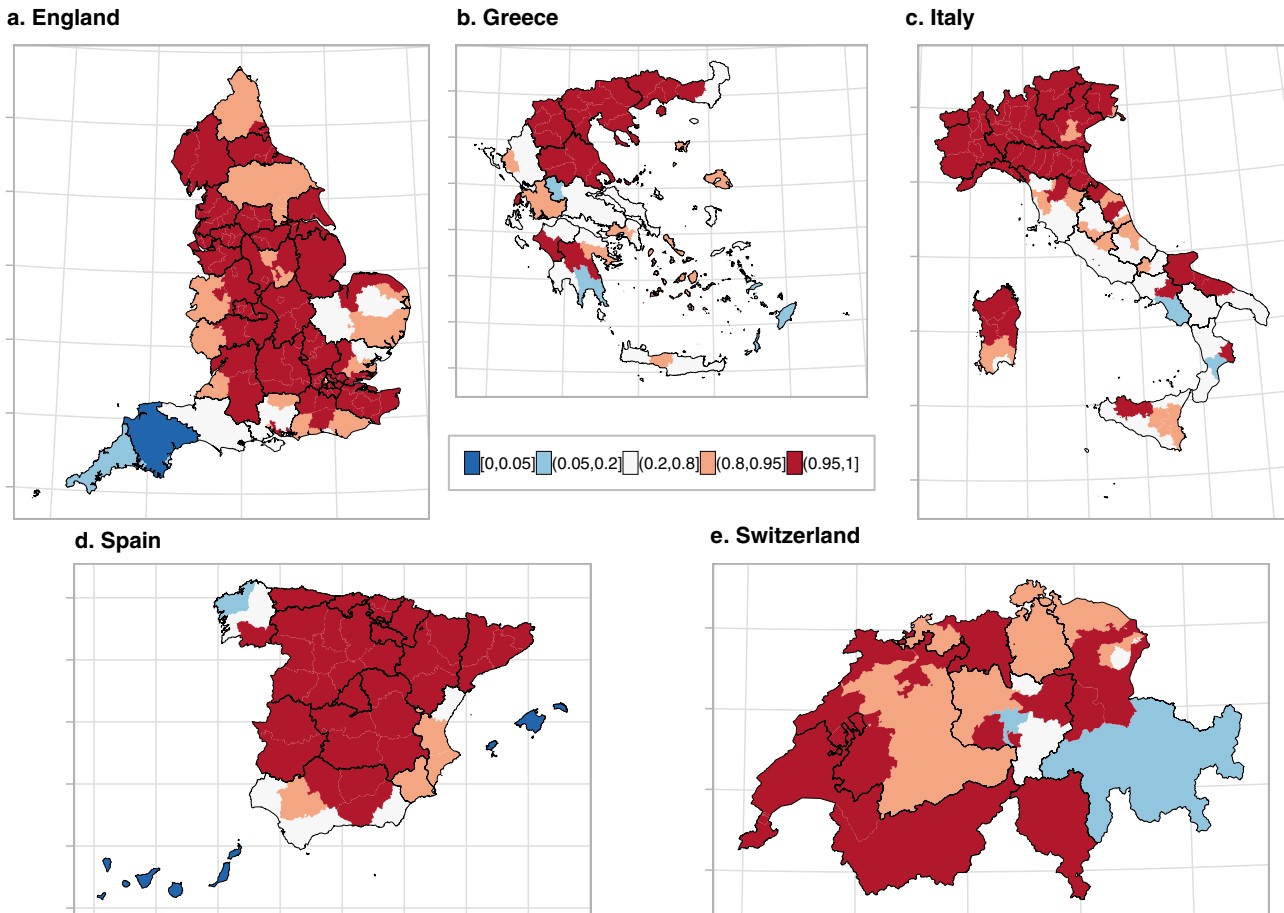

**Fig. 3 Probability that the relative excess deaths is higher than 0% by NUTS3 region in 2020. a** Probability that the relative excess deaths is higher than 0% in England. **b** Probability that the relative excess deaths is higher than 0% in Greece. **c** Probability that the relative excess deaths is higher than 0% in Italy. **d** Probability that the relative excess deaths is higher than 0% in Spain and **e** probability that the relative excess deaths is higher than 0% in Switzerland. Areas in blue indicate areas that observed less deaths than expected had the pandemic not occurred, whereas the different shades of red indicate the higher relative excess mortality. The black solid lines correspond to the NUTS2 region borders.

latter reported a 15.6% excess for 2020 compared to the average number of deaths 2015 to 2019[26]. In Spain, the relative excess mortality varied from 26.8% to 77.9% across the different age groups for the period March to May 2020 and from 10.0% to 18.9% during the period July to December 2020[27].

At the regional level, our findings align with the Office for National Statistics in England, which reported a 20% increase in the relative excess mortality in London, the largest relative excess observed nationwide in 2020[11]. Our estimates are also in line with the Hellenic Statistical Authority reports suggesting that Macedonia and Thrace experience the largest relative increase in excess deaths in Greece (14.9% in males and 12.9% in females)[10]. The observed north-to-south geographical gradient in the impact of COVID-19 in Italy is in line with previous studies[8,9]. The Italian National Institute of Statistics reported that the provinces with the highest excess of mortality in Northern Italy were Bergamo (51.5%), Cremona (47.5%), Lodi (39.9%) and Piacenza (35.7%)[26]. In central and southern Italy, Pesaro-Urbino (21.1%) and Foggia (16.1%) were the most affected provinces[26]. The Instituto de Salud Carlos III together with the European mortality monitoring initiative (EuroMOMO) reported the highest excess mortality in Madrid, varying from 17.6% for age group 65 to 74 years during August 2020 to January 2021 to 21.6% for those aged 74 years or older during the period March to May 2020[27]. Similarly to our study, Madrid, Castile-La Mancha, Castile-Leon, and Catalonia had the highest excess mortality[27].

Selected studies focusing on regional excess mortality during 2020 in countries not included in the current study have also reported geographical discrepancies[23,28]. A study focusing in selected Latin America countries found that the subregions of Roriama, Lima, Puebla and Santa Elena were the subregions most affected in Brazil, Peru, Mexico and Ecuador with the percentage increase in the excess mortality varying from 50 to 160% during 2020[28]. These values are much higher compared to the maximum observed increase in excess mortality in our study which was 33% increase in males across all ages in Madrid. Similarly, a lot of geographical variation was observed in the US with the maximum percentage increase in excess mortality observed in the state of New Jersey during 2020, with the excess being approximately 35%, comparable with the value observed in males in Madrid[23]. Regional studies focusing on the early stages of the pandemic reported geographical discrepancies in Europe and Asia. A study in France reported 63.7% excess mortality in Île-de-France during March and May 2020[29]. A study focusing on the three months of the COVID-19 outbreak in China reported 56% (33%–87%) increase in the excess in three Wuhan districts, whereas no significant excess in the rest of China[30].

Several factors may have contributed to the differences in excess mortality we observed during the COVID-19 pandemic across countries and regions. Mortality depends on the probability of being infected and mortality among those infected. Both probabilities vary depending on the country's demographic and

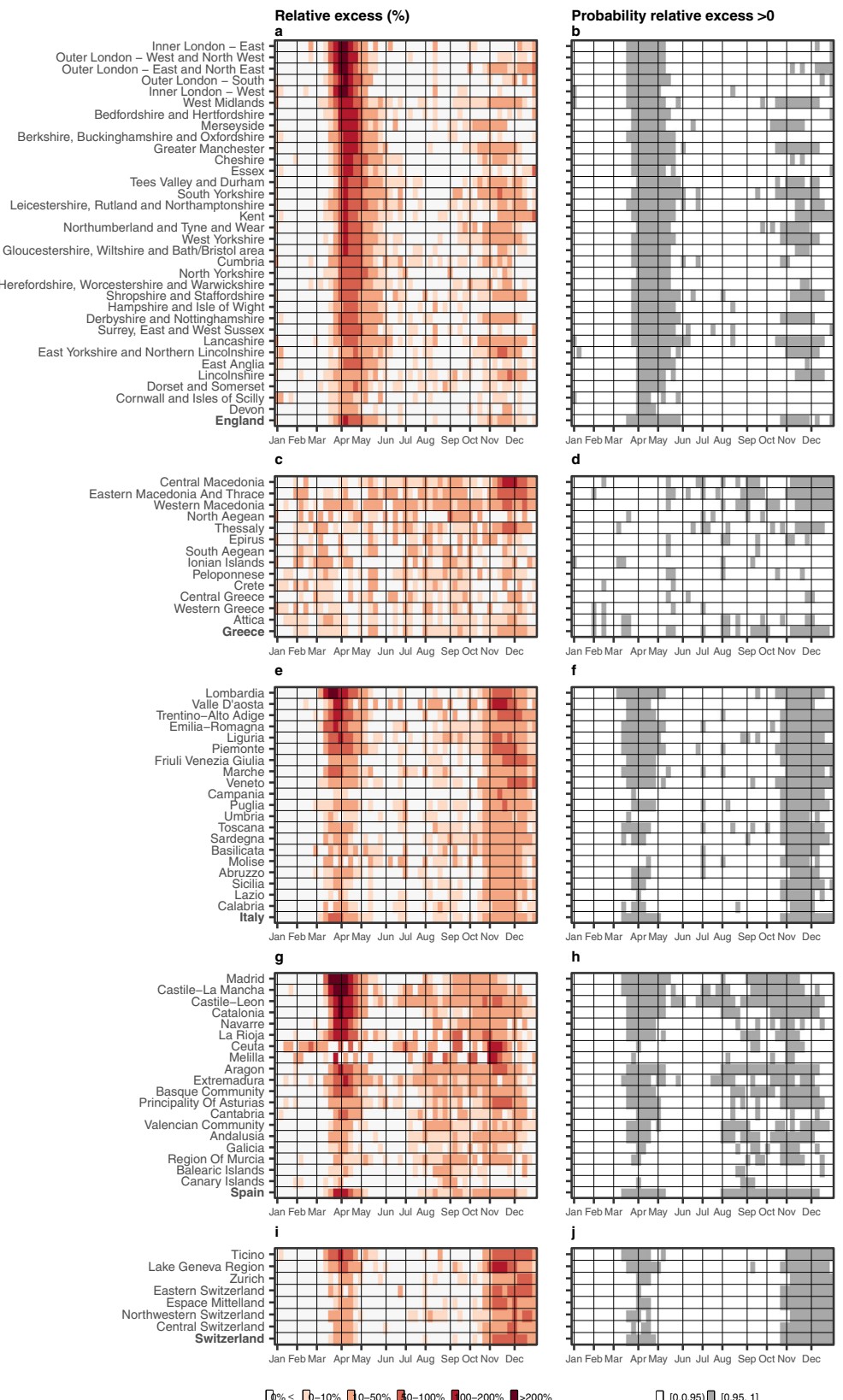

socio-economic characteristics, including age structure, ethnicity, level of deprivation, and environmental factors[14]. Further, the timeframe of non-pharmaceutical interventions in countries and regions and the resilience and capacity of health care systems have played a role[7]. The mobility of populations across borders and between regions and the timeliness of lockdowns have probably been the most important factors[31,32]. We observed weak, but consistent, evidence of differences in the excess mortality by sex in England, with potential explanations being risk factors that are known to change with sex including differences in occupation, lifestyle, medical comorbidities, or use of medications[33].

**Fig. 4 Weekly median relative excess deaths (%) across the different countries by NUTS2 region in 2020 (left) and corresponding probability that the weekly relative excess is larger than 0% (right). a** Weekly median relative excess deaths in England. **b** Posterior probability that the relative excess is higher than 0% in England. **c** Weekly median relative excess deaths in Greece. **d** Posterior probability that the relative excess is higher than 0% in Greece. **e** Weekly median relative excess deaths in Italy. **f** Posterior probability that the relative excess is higher than 0% in Italy. **g** Weekly median relative excess deaths in Spain. **h** Posterior probability that the relative excess is higher than 0% in Spain. **i** Weekly median relative excess deaths in Switzerland and **j** posterior probability that the relative excess is higher than 0% in Switzerland. Different shades of red on the panels **a**, **c**, **e**, **g** and **i** indicate higher relative excess mortality, whereas the white ones relative excess mortality lower than 0%. The white colour on the panels **b**, **d**, **f**, **h** and **j** indicate insufficient evidence of a relative excess larger than 0%, whereas the grey strong evidence.

The first wave of the pandemic was mainly exogenous, with international airports and transport routes serving as main entry points. Thus, the highest number of excess deaths during the first wave was observed in the areas affected first, i.e., big transit hubs like London, Madrid, Lombardia and Ticino, and Geneva. From the initial point of introduction, SARS-CoV-2 spread to nearby large urban areas where community transmission was established and increased exponentially, spreading to the entire country in the absence of mobility restrictions[34]. Furthermore, during the first wave stochastic super-spreader events like the Champion's League football game between Atalanta and Valencia on February 19, 2020[35] played an important role in establishing community transmission[36]. The lockdowns in Italy, England and Spain were introduced after community transmission was established in the areas first affected. On the day of the national lockdown 1,797 new cases were reported in Italy, 1,159 in Spain and 2,349 in the UK. The lockdown reduced mobility, allowing some areas to maintain lower levels of community transmission. The importance of the timeliness of the lockdown is, among other results, highlighted in the example of Italy, where the large number of COVID-19 cases in the North forced nationwide mobility restrictions, benefiting the south of Italy where community transmission was not established, creating a North to South gradient in the excess mortality[31]. In Greece, a timely nationwide lockdown was imposed on March 13, 2020, before the country reached 100 reported cases per day, potentially explaining the lack of excess mortality nationwide during the first six months of 2020. However, given the data available, it is not trivial to disentangle the specific effect of lockdown measures from other control measures and changes in behaviour that occurred simultaneously in response to the pandemic such as spontaneous social distancing, face mask usage, or contact tracing.

The spatial distribution of excess mortality during the second wave of the pandemic was more homogeneous, reflecting multiple routes of entry and transmission. Factors contributing to this situation included the relaxation of non-pharmaceutical interventions with the reopening of schools, retail and other activities, domestic and international travel, and the public's loosening of preventive behaviours[37]. The timeliness of the lockdowns and population mobility again played a crucial role. Lockdowns during the second wave were slower to be implemented and less rigorous[38]. In Italy and Switzerland, the geographical distribution of the excess deaths was equal nationwide, whereas it was more variable in England, Greece and Spain. In Greece, where community transmission was not established during the first epidemic wave, the patterns observed were highly localised, mimicking the patters observed in the other countries during the first epidemic wave. Central Macedonia (with the transit hub Thessaloniki), Eastern Macedonia and Thrace, and Western Macedonia which border on Bulgaria and Turkey, are the hardest-hit regions. On November 3, 2020, the Greek nationwide lockdown limited transmission in the rest of the country, resulting in lower excess mortality in areas in the south. In Switzerland, the area hit hardest during the second wave was the lake of Geneva region, potentially influenced by the French second wave[37].

In conclusion, this study provides the first comprehensive analysis of weekly sub-national excess mortality for 2020 across five countries, disaggregated by sex and age groups. Our findings highlight how excess mortality varied largely across countries, within countries and over time. They suggest that a timely lockdown led to reduced community transmissions and, subsequently, lower excess mortality. However, lockdowns have adverse short and long-term health, psychosocial and economic effects that need to be considered[7,39]. Community transmission was established in the transit hubs and nearby large metropolitan areas during the first stages of the pandemic. Therefore, rapid action to limit transmission around these hubs is essential to prevent spread to other regions and countries.

## Methods

**Ethics.** The analyses for England were covered by national research ethics approval from the London-South East Research Ethics Committee (Reference 17/LO/0846). Data access was covered by the Health Research Authority Confidentiality Advisory Group under section 251 of the National Health Service Act 2006 and the Health Service (Control of Patient Information) Regulations 2002 (Reference 20/CAG/0028). Data access for Greece was approved by the 28th Statistical Privacy Committee and the relevant letter of the president of the Hellenic Statistical Authority (ΓΠ-498/23-12-2020) in accordance with section 5.3 of the Minutes of the Statistical Privacy Committee. The study is about secondary, aggregate anonymised data so no additional ethical permission is required.

**All-cause mortality.** We retrieved data for all-cause deaths and population counts from the Office for National Statistics in England (derived from the national mortality and birth registrations and the Census), the Hellenic Statistical Authority in Greece, the Italian National Institute of Statistics in Italy, the National Centre of Epidemiology at the Carlos III Health Institute and the Daily Monitoring Mortality System and also the National Statistics Institute and Ministry of Justice in Spain and the Federal Statistical Office in Switzerland, Supplementary Table 8. We selected the current Nomenclature of Territorial Units for Statistics (NUTS) and in particular NUTS3 (small regions for specific diagnoses) as the main spatial unit of our analysis[40]. We also show results at the NUTS2 level, which is defined to reflect basic regions for the application of regional policies (https://ec.europa.eu/eurostat/web/nuts/background/). The number of deaths from all-causes and the population denominator was available by sex, age, week and NUTS3 region defined as areas with a population varying from 150,000 to 800,000, for 2015-2020. We used the International Organization for Standardization (ISO) week calendar, i.e. the seven consecutive days beginning with a Monday and ending with a Sunday. We aggregated mortality and population data by age groups <40, 40–59, 60–69, 70–79 and 80 years and above to maintain consistency between countries and the literature[12].

**Population at risk.** Population estimates for the years 2014–2020 are available for Greece, Italy and Spain for the January 1 of every year, whereas for Switzerland for December 31, Supplementary Table 7. To obtain weekly 2020 population Figures we performed a two-step linear interpolation. In a first step, using the years 2015–2020, we predicted population counts by age, sex and NUTS3 regions for January 1, 2021. In a second step, we calculated weekly 2020 population Figures by linear interpolation of the estimates on January 1, 2020, and January 1, 2021, by age, sex and NUTS3 regions. For England, mid-year population Figures were available, Supplementary Table 7, which for 2020 were affected by COVID-19 deaths during the first wave. We, therefore, used the data for 2015 to 2019 and estimated the midyear population of 2020 through linear interpolation for England. We estimated population numbers for January 1 2020, and then used linear interpolation to obtain the weekly population of 2019, which we used as a proxy for 2020.

**Covariates**. As ambient temperature influences death rates[41], we retrieved data on temperature from the ERA5 reanalysis data set of the Copernicus climate data[42]. Using data from global in situ and satellite measurements, ERA5 provides hourly estimates of a large number of atmospheric, land and oceanic climate variables, spatially and temporally compatible with our analysis[42]. For each centroid of the grid cells (at $0.25° \times 0.25°$ resolution) that fall into the NUTS3 regions, we calculated the daily mean temperature during 2015–2020 and then the weekly mean, align temperature and mortality data. We are modelling weekly deaths due to data availability, hence we are implicitly considering a 0-7 lag. Additionally, as mortality from all causes can be different during national holidays, we also included a binary variable taking the value 1 if the week contains a public holiday and 0 otherwise.

**Statistical methods**. We used Bayesian hierarchical models to predict deaths in 2020, under the scenario of absence of the pandemic. Let $y_{jtsk}$ be the number of all-cause deaths, $\boldsymbol{P}_{jtsk}$ be the population at risk and $r_{jtsk}$ the risk in the $j$-th week of the $t$-th year ($t = 1, …, 5$ with year 1 corresponding to 2015), for the $s$-th spatial unit ($s = 1, …, S$) and $k$-th age-sex group ($k = 1, …, 10$) (male-female and <40, 40–59, 60–69, 70–79, ≥80). The models in the main analysis excluded the age group below 40 years, based on the cross-validation exercise. We assume a Poisson distribution for the number of deaths $y_{jtsk}$ and modelled the risk $r_{jtsk}$ using the following specification:

$$y_{jtsk} \sim \text{Poisson}\left(r_{jtsk}\boldsymbol{P}_{jtsk}\right)$$
$$\log\left(r_{jtsk}\right) = \beta_{0t} + \beta_1 Z_j + \text{f}\left(x_{jts}\right) + b_s + w_j,$$

where $\beta_{0t}$ is the year specific intercept given by $\beta_{0t} = \beta_0 + \epsilon_t$, with $\beta_0$ being the global intercept and $\epsilon_t \sim \text{Normal}\left(0, \tau_\epsilon^{-1}\right)$ an unstructured random effect representing the deviation of each year from the global intercept, with $\tau_\epsilon$ denoting the precision of $\epsilon_t$. The term $\beta_1$ represents the effect of public holidays (i.e., $Z_j = 1$ if week $j$ contains a public holiday and 0 otherwise). The linear predictor includes also a non-linear effect f(·) of the average weekly temperature in each area, $x_{jts}$; in particular, we assume the following second-order random walk (RW2) model:

$$x_{jts}|x_{(j-1)ts}, x_{(j-2)ts}, \tau_x \sim \text{Normal}\left(2x_{(j-1)ts} + x_{(j-2)ts}, \tau_x^{-1}\right),$$

with $\tau_x$ denoting the precision.

The term $b_s$ is a spatial field defined as an extension of the Besag-York-Mollié model given by the sum of an unstructured random effect, $v_s \sim \text{Normal}\left(0, \tau_v^{-1}\right)$, and a spatially structured effect $u_s$[43–45]. In particular $b_s$ is defined as follows:

$$b_s = \frac{1}{\sqrt{\tau_b}}\left(\sqrt{1-\phi}v_s^\star + \sqrt{\phi}u_s^\star\right),$$

where $u_s^\star$ and $v_s^\star$ are standardised version of $u_s$ and $v_s$ to have variance equal to 1[46]. The term $0 \leq \phi \leq 1$ is a mixing parameter which measures the proportion of the marginal variance explained by the structured effect.

To account for seasonality, we included in the linear predictor a non-linear weekly effect $w_j$, common to all the areas, with a first order random walk (RW1) structure:

$$w_j|w_{j-1}, \tau_w \sim \text{Normal}\left(w_{j-1}, \tau_w^{-1}\right),$$

where $\tau_w$ is the precision of $w_j$.

We specified minimally informative prior distributions, i.e. for the fixed effects $\beta_0$ and $\beta_1$. For the spatial field hyperparameters $\phi$ and $\tau_b$ we adopted priors that tend to regularise inference while not providing too strong information, the so-called penalize complexity (PC) priors introduced in[46]. In particular, for the standard deviation $\sigma_b = \sqrt{1/\tau_b}$ we selected a prior so that $\Pr(\sigma_b > 1) = 0.01$, implying that it is unlikely to have a spatial relative risk higher than exp(2) based solely on spatial or temporal variation. For $\phi$ we set $\Pr(\phi < 0.5) = 0.5$ reflecting our lack of knowledge about which spatial component, the unstructured or structured, should dominate the field $b$. Finally, PC priors are also adopted for all the standard deviations $\sigma_\epsilon = \sqrt{1/\tau_\epsilon}$, $\sigma_x = \sqrt{1/\tau_x}$ and $\sigma_w = \sqrt{1/\tau_w}$ such that for each hyperparameter $\Pr(\sigma > 1) = 0.01$.

We train the model using the years 2015–2019 and predict area level weekly mortality for 2020 assuming that the pandemic did not take place. To summarise the results we retrieve samples by age, sex, week and NUTS3 regions for 2020 from the posterior predictive distribution:

$$p(y_{jsk6}|\mathbf{D}) = \int p(y_{jsk6}|\boldsymbol{\theta})p(\boldsymbol{\theta}|\mathbf{D})d\boldsymbol{\theta},$$

where $\boldsymbol{\theta}$ is the vector of the model parameters and $\mathbf{D}$ the observed data. We report the weekly observed number of deaths in 2020 together with $p(y_{jsk6}|\mathbf{D})$ at the weekly resolution. We also compare $p(y_{jsk6}|\mathbf{D})$ with the observed number of deaths in 2020 and retrieve the posterior of the relative (percent) increase in mortality (i.e. relative to what is expected had the pandemic not occurred). We summarise the above posterior reporting medians, 95% credible intervals and posterior probability that the relative excess mortality is larger than 0. We also present half-year estimates of the median excess mortality by NUTS3 regions, age and sex and the posterior probability that the excess is larger than 0.NUTS3 regions and weeks were

the main spatial and temporal resolution of the analysis. To calculate the different temporal, spatial, sex and age specific aggregations, we combined $p(y_{jsk6}|\mathbf{D})$ for the different age and sex groups or integrate in time or space accordingly, resulting in the corresponding posteriors.

**Model validation**. We perform a cross-validation like procedure to examine the validity of our predictions. Using the years 2015–2019, we fit the proposed model multiple times, leaving out one year at a time and predicting the weekly number of deaths by NUTS3 regions for the year left out. We repeat for the different age and sex groups, and different countries. We assess the agreement between the predicted and observed deaths at the year $t$. We use the following metrics: a) the correlation between the predicted and observed deaths and b) the 95% coverage, defined as the probability that the observed deaths lie within the 95% interval estimated from the model.

**Reporting summary**. Further information on research design is available in the Nature Research Reporting Summary linked to this article.

## Data availability

We provide at https://github.com/gkonstantinoudis/ExcessDeathsCOVID the final version of the datasets for Italy and Switzerland as their mortality data is available online. Raw mortality data files for Switzerland are provided at https://www.bfs.admin.ch/bfs/en/home/statistics/population/births-deaths/deaths.assetdetail.19184461.html and https://www.bfs.admin.ch/bfs/en/home/statistics/population/births-deaths/deaths.assetdetail.13187299.html, whereas for Italy at https://www.istat.it/it/archivio/240401. Access to mortality data for Greece, England and Spain is subject to requests. For England, the data were obtained from the Small Area Health Statistics Unit (SAHSU), which does not have permission to supply data to third parties. The data can be requested through the Office for National Statistics (https://www.ons.gov.uk/). For Greece, mortality data can be requested from ELSTAT (https://www.statistics.gr) and for Spain from the National Centre of Epidemiology at the Carlos III Health Institute (https://eng.isciii.es/eng.isciii.es/Paginas/Inicio.html).

Population data is available at the following locations:

England: https://www.ons.gov.uk/peoplepopulationandcommunity/populationandmigration/populationestimates/datasets/populationestimatesforukenglandandwalesscotlandandnorthernireland

Greece: The selected aggregation by age, sex and NUTS3 regions is subject to a request at: https://www.statistics.gr/en/statistical-data-request

Italy: http://demo.istat.it/ricostruzione/download.php?lingua=ita for 2015–2019 and http://demo.istat.it/popres/download.php?anno=2020&lingua=itafor2020

Spain: https://www.ine.es/jaxiT3/Tabla.htm?t=9691

Switzerland: https://www.pxweb.bfs.admin.ch/pxweb/en/px-x-0102010000_102/-/px-x-0102010000_102.px/

Air temperature at 2m for all countries was retrieved from https://cds.climate.copernicus.eu/cdsapp#!/dataset/reanalysis-era5-land?tab=form.

## Code availability

All models were fitted using the Integrated Nested Laplace Approximation (INLA) using its R software interface[47]. To ensure reproducibility and transparency to our results and approach the code for running the analysis is available online[48]. Results are also provided in a Shiny app (http://atlasmortalidad.uclm.es/excess/), to facilitate communication with the general public and stakeholders.

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

## Acknowledgements

We thank Hima Daby, Gajanan Natu and Eric Johnson for their help with data acquisition, storage, preparation and governance. We also thank the University of Castilla-La Mancha for hosting the shiny server. All authors acknowledge Infrastructure support for the Department of Epidemiology and Biostatistics provided by the NIHR Imperial Biomedical Research Centre (BRC). G.K. is supported by an MRC Skills Development Fellowship [MR/T025352/1]. M.B. is supported by a National Institutes of Health, grant number [R01HD092580-01A1]. Infrastructure support for this research was provided by the National Institute for Health Research Imperial Biomedical Research Centre (BRC). The work was partly supported by the MRC Centre for Environment and Health, which is funded by the Medical Research Council (MR/S019669/1, 2019-2024). V. Gómez-Rubio is supported by grant SBPLY/17/180501/000491, funded by Consejería de Educación, Cultura y Deportes (JCCM, Spain) and FEDER, and grant PID2019-106341GB-I00, funded by Ministerio de Ciencia e Innovación (Spain). M.E. and J.R. are supported by by the Swiss National Science Foundation (grant 189498). The study uses Small Area Health Statistics (SAHSU) data, obtained from the Office for National Statistics. The work of the UK SAHSU Unit is overseen by Public Health England (PHE) and funded by PHE as part of the MRC-PHE Centre for Environment and Health also supported by the UK Medical Research Council, Grant number: MR/L01341X/1), and the National Institute for Health Research (NIHR) through its Health Protection Units (HPRUs) at Imperial College London in Environmental Exposures and Health and in Chemical and Radiation Threats and Hazards, and through Health Data Research UK (HDR UK). This paper does not necessarily reflect the views of Public Health England, the National Institute for Health Research or the Department of Health and Social Care.

## Author contributions

G.K. and M.B. conceived the study. M.B. supervised the study. G.K. developed the initial study protocol and discussed it with M.B., M.C., M.P. and G.B. G.K. developed the statistical model, prepared the population and covariate data and led the acquisition of mortality data. M.C. validated and modified accordingly the code. G.K. ran the analysis for England, Greece and Switzerland, M.C. for Italy and I.L.G for Spain. G.K. wrote the initial draft and all the authors contributed in modifying the paper and critically interpreting the results. P.V, J.R., M.E. and A.L. commented extensively on previous drafts. V.G.R. developed the Shiny app. All authors read and approved the final version for publication.

## Competing interests

The authors declare no competing interests.
