## [Peer Review File · Nature Communications]

Regional excess mortality during the 2020 COVID-19 pandemic in five European countriesREVIEWER COMMENTS

Reviewer #1 (Remarks to the Author):

This study makes a welcome contribution to comparative research on the consequences of the COVID-19 pandemic in terms of elevated mortality during 2020, in a set of countries in Europe. Its main contribution is that it provides high quality information on the spatial as well as the temporal dimensions of the excess mortality during the pandemic, and the intersection of these two dimensions. For comparative research, it makes good sense to focus on excess mortality rather than cause of death-specific statistics as the systems for registration of causes of death may differ across countries. I think the study would benefit from just a few clarifications and modifications of the presentation. I suggest two main issues and a few minor ones to consider.

1) The Bayesian model that is used to estimate the degree of excess mortality in different regions and across time appears to produce very efficient and precise estimates that also account for the long-term trends in terms of annual mortality reductions (cf. Figures S2-S6). Many other studies have relied on much more crude estimates where the death counts in 2020 are simply compared to an average of the death counts during a set of pre-pandemic years. The latter approach produces estimates of excess mortality that are too low to be entirely realistic. I think the authors could help guide the reader in these matters and high-light the advantages of their Bayesian approach. However, I must admit that I am a bit surprised to read that the authors find that their own estimates are still lower than those in other previous statistics on the topic and wonder whether I missed something.

2) Clearly, the key contribution of this submission is the presentation of the combination of the spatial and temporal dimensions of the excess mortality during the pandemic, as in Figure 4. I think this could be high-lighted a bit better and would encourage the authors to present half-year versions of the maps on the regional patterns in excess mortality during 2020, as in Figures S7-S16. I realize that there are many maps already, but instead of two versions of excess mortality we could be provided two temporal versions based on just one of these two versions.

A few additional issues could also be addressed:

- The authors discuss the indirect contributions of non-COVID-19 deaths as mainly contributing to more deaths. I think recent evidence suggests that the indirect effects during the pandemic year have mainly worked in the opposite direction.
- Line 63 of the text refers to population numbers in 2020; please specify which part of that year that is considered.
- Lines 65-68 refers to death counts in relation to those of a reference period of five years prior to the pandemic. It is unclear if the comparison in Table 1 is based on the crude death counts in that five-year period or something produced by the modelling the authors carry out. If referring to crude statistics, see my comment 1 above.
- Page 4 contains section headings that refer to Sub-national level trends and finer such trends. However, the maps that are presented provide no trends, so "patterns" would make a better label.
- The authors claim that their findings suggest that timely lock-downs have led to reduced community transmissions. The authors could elaborate a bit more on how they reached such a conclusion.

mvh Gunnar Andersson

Reviewer #2 (Remarks to the Author):

The manuscript "Regional excess mortality during the 2020 COVID-19 pandemic: a study of five European countries" by Konstantinou et al. presents age and sex specific excess deaths in 2020 in five European countries at different spatial resolutions. They have shown how excess mortality varied among countries, within each of one the five countries and through time using a hierarchical Bayesian spatio-temporal model. I enjoyed reading the manuscript, I believe the chosen method is adequate to predict the number of deaths in 2020 if COVID-19 did not happen. The authors provide satisfactory explanations/hypothesis for the spatial and spatio-temporal heterogeneities in excess deaths.

Some comments:

Sex differences: When comparing the excess deaths in males and females in each country (lines 65 to 68) the figures were similar in Spain, Italy, Greece and Switzerland, however the difference in England is quite large, the excess mortality in men is much higher than in women. Do the authors have any explanation for that? Looking at tables S2-S6 most of the median relative excess deaths in men are higher than in women, the majority of the intervals overlap though. In figure S2 we see that the expected number of deaths per week in men is already greater than in women (excepted for people older than 80 y.o.). But the observed deaths between March and June is also higher for men and nearly similar for people older than 80. This behaviour is similar in other countries (Figures S3-S6), so I wonder why male deaths in England was so different (line 66), perhaps the credible intervals overlap but still a quite large difference.

Modelling: The authors assumed that for each country the number of deaths over time (week and year), space and age-sex group follows a Poisson model. That is OK, perhaps the author could try a negative binomial but that is not an issue because that random effects may take into account the overdispersion. What was spatial level used? Did the author fit three models per country (National, NUTS2 and NUTS3) and for each model use the forecasted values for 2020, or just one model at NUTS3 level and the NUTS2 and National level statistics were calculated by aggregating the forecasted values for 2020? These two approaches do not lead to the same results for NUTS2 and national level.

Temperature: In the proposed model the authors have added temperature as a covariate in a spline-like fashion by using a RW2 model, what was the temperature effect on mortality risk? Is there any? Should the temperature be lagged? It would be nice some comments on this issue. Is winter worst for elder people?

Population: The population projections were used essentially as offset in the fitted models, however there are some uncertainty on these projections the further from the last census the bigger the uncertainty. Hence the credible intervals for the mortality risk should be larger if the population uncertainty were propagated in the model. I am not asking to do this, but add as a limitation. There are ways to estimated projected populations in a Bayesian fashion see Raftery et al. (2014) and Gerland et al. (2014)

Excess deaths in other countries/regions/cities: The authors have avoided to discuss excess mortality from countries other than the five chosen ones. I quite understand why, however, it would be interesting in the discussion some values for excess mortality from other countries like the US, Brazil, Mexico, Japan, Korea, France, Sweden, etc. to illustrate how homogeneous (or heterogeneous) are the studied countries, how large or small were the excess deaths in space and time for instance in Madrid compared to Sao Paulo. I am not asking a systematic review, nor the values for all these places, but it would be interesting to see some in order to understand how are the five studied countries compared to other places (in Europe and/or throughout the world).

References:

Raftery et al. (2014) <https://pubmed.ncbi.nlm.nih.gov/25324591/>

Gerland et al. (2014) <https://www.science.org/doi/10.1126/science.1257469>

NCOMMS-21-37776

Regional excess mortality during the 2020 COVID-19 pandemic: a study of five European countries

We are very grateful to the reviewers for helping to improve the manuscript. See below a detailed response to the reviewers' comments. We have also uploaded two versions of the revised manuscript, one with the changes marked (in blue) and a clean one. In addition, we have modified the data availability statement to include detailed information with the exact link of the data sources and the figures and captions in compliance with the Nature Communications requirements.

Reviewer Comments

Reviewer #1

This study makes a welcome contribution to comparative research on the consequences of the COVID-19 pandemic in terms of elevated mortality during 2020, in a set of countries in Europe. Its main contribution is that it provides high quality information on the spatial as well as the temporal dimensions of the excess mortality during the pandemic, and the intersection of these two dimensions. For comparative research, it makes good sense to focus on excess mortality rather than cause of death-specific statistics as the systems for registration of causes of death may differ across countries. I think the study would benefit from just a few clarifications and modifications of the presentation. I suggest two main issues and a few minor ones to consider.

1) The Bayesian model that is used to estimate the degree of excess mortality in different regions and across time appears to produce very efficient and precise estimates that also account for the long-term trends in terms of annual mortality reductions (cf. Figures S2-S6). Many other studies have relied on much more crude estimates where the death counts in 2020 are simply compared to an average of the death counts during a set of pre-pandemic years. The latter approach produces estimates of excess mortality that are too low to be entirely realistic. I think the authors could help guide the reader in these matters and high-light the advantages of their Bayesian approach. However, I must admit that I am a bit surprised to read that the authors find that their own estimates are still lower than those in other previous statistics on the topic and wonder whether I missed something.

Authors' reply: We thank the reviewer for this remark. We have now added a couple of sentences to clarify the advantages of our approach in comparison with other non-model based studies, page 6 and lines 147-150:

Our modelling approach accounts for spatial and temporal mortality trends, factors such as temperature and public holidays, and the different population temporal trends across space, age and sex groups, factors not taken into account in previous reports [1, 2].

Our estimates are lower in comparison with previous studies, but when the comparison is made accounting for the uncertainty in the estimates, the differences are relatively small. As the mortality dynamics are dependent on temperature trends, seasonality and population trends, it is hard to predict the direction of the effect. To clarify further the discrepancies observed we rephrased on page 6 and lines 160-164:

Our point estimates are lower but when we consider their uncertainty, the differences are relatively small

and could be explained by the periods used to train the model, the data sources used, the prediction periods and the different population estimation approaches [3]. Other data-related differences may play a role: for instance, previous analyses considered England and Wales [4], while our focus is on England only.

2) Clearly, the key contribution of this submission is the presentation of the combination of the spatial and temporal dimensions of the excess mortality during the pandemic, as in Figure 4. I think this could be high-lighted a bit better and would encourage the authors to present half-year versions of the maps on the regional patterns in excess mortality during 2020, as in Figures S7-S16. I realize that there are many maps already, but instead of two versions of excess mortality we could be provided two temporal versions based on just one of these two versions.

Authors' reply: All the requested maps are now added on the Online Supplement, see Supplementary Figures S22-41. We also added the following sentences in the main manuscript referring to this new addition, page 5, lines 125-129:

Supplementary Fig. S22-41 show the half-year median excess mortality by age, sex and NUTS3 regions and the posterior probability that the excess is larger than 0. Across the five countries the north of Italy and central Spain were hit the most during the first 6 months of 2020. Greece was mainly affected during the second 6 months of the pandemic with the North of the country and the elderly most affected, Supplementary Figures S26-29.

page 8, lines 219-221:

In Greece, a timely nationwide lockdown was imposed on March 13, 2020, before the country reached 100 reported cases per day, potentially explaining the lack of excess mortality nationwide during the first six months of 2020.

and page 12, lines 311-313:

We also present half-year estimates of the median excess mortality by NUTS3 regions, age and sex and the posterior probability that the excess is larger than 0.

A few additional issues could also be addressed:

- The authors discuss the indirect contributions of non-COVID-19 deaths as mainly contributing to more deaths. I think recent evidence suggests that the indirect effects during the pandemic year have mainly worked in the opposite direction.

Authors' reply: We thank the reviewer for this comment. We have now added a couple of sentences about this in the discussion, page 5-6, lines 144-147:

In contrast, a population-based study in the UK found that that primary care contacts for physical and mental health conditions decreased after the introduction of lockdowns in March, 2020, and remained below pre-lockdown levels [5]. These trends, could represent a substantial burden of unmet needs, with potential implications for subsequent morbidity and premature mortality [5].

- Line 63 of the text refers to population numbers in 2020; please specify which part of that year that is considered.

Authors' reply: The population number refer to the 1st of January in 2020 for all countries. Table S8 also shows the time point for which population data were available in the different countries. Notice that the estimates for this table were retrieved based on the population interpolation and were rounded. We have now rephrased accordingly on the caption of Table 1:

The population numbers refer to the 1st of January in 2020.

- Lines 65-68 refers to death counts in relation to those of a reference period of five years prior to the pandemic. It is unclear if the comparison in Table 1 is based on the crude death counts in that five-year period or something produced by the modelling the authors carry out. If referring to crude statics, see my comment 1 above.

Authors' reply: We thank the reviewer for this comment. We now clarify on the caption of Table 1:

The expected number of deaths were calculated as the mean of crude deaths by age and sex during 2015-2019.

- Page 4 contains section headings that refer to Sub-national level trends and finer such trends. However, the maps that are presented provide no trends, so "patterns" would make a better label.

Authors' reply: We thank the reviewer for pointing that out. We have revised accordingly on page 4.

- The authors claim that their findings suggest that timely lock-downs have led to reduced community transmissions. The authors could elaborate a bit more on how they reached such a conclusion.

Authors' reply: We thank the reviewer for giving us the opportunity to further clarify. This conclusion is supported by the generic geographical patterns and strongly supported by the results from Italy and Greece. We rephrased accordingly for further clarification on page 8, lines 216-224:

The importance of the timeliness of the lockdown is, among other results, highlighted in the example of Italy, where the large number of COVID-19 cases in the North forced nationwide mobility restrictions, benefiting the south of Italy where community transmission was not established, creating a North to South gradient in the excess mortality [6]. In Greece, a timely nationwide lockdown was imposed on March 13, 2020, before the country reached 100 reported cases per day, potentially explaining the lack of excess mortality nationwide during the first six months of 2020. However, given the data available, it is not trivial to disentangle the specific effect of lockdown measures from other control measures and changes in behaviour that occurred simultaneously in response to the pandemic such as spontaneous social distancing, face mask usage, or contact tracing.

Reviewer #2

The manuscript "Regional excess mortality during the 2020 COVID-19 pandemic: a study of five European countries" by Konstantinou et al. presents age and sex specific excess deaths in 2020 in five European countries at different spatial resolutions. They have shown how excess mortality varied among countries, within each of one the five countries and through time using a hierarchical Bayesian spatio-temporal model. I enjoyed reading the manuscript, I believe the chosen method is adequate to predict the number of deaths in 2020 if COVID-19 did not happen. The authors provide satisfactory explanations/hypothesis for the spatial and spatio-temporal heterogeneities in excess deaths.

Some comments:

Sex differences: When comparing the excess deaths in males and females in each country (lines 65 to 68) the figures were similar in Spain, Italy, Greece and Switzerland, however the difference in England is quite large, the excess mortality in men is much higher than in women. Do the authors have any explanation for that? Looking at tables S2-S6 most of the median relative excess deaths in men are higher than in women, the majority of the intervals overlap though. In figure S2 we see that the expected number of deaths per week in men is already greater than in women (excepted for people older than 80 y.o.). But the observed deaths between March and June is also higher for men and nearly similar for people older than 80. This behaviour is similar in other countries (Figures S3-S6), so I wonder why male deaths in England was so different (line 66), perhaps the credible intervals overlap but still a quite large difference.

Authors' reply: We thank the reviewer for observing this and giving us the opportunity to clarify, on page 4 and lines 102-103:

Overall, the estimates of the excess mortality in males and females were similar, with the excess being consistently lower for females in England, Supplementary Table S2 and Fig. 1.

And page 7, lines 202-205:

We observed weak, but consistent, evidence of differences in the excess mortality by sex in England, with potential explanations being risk factors that are known to change with sex including differences in occupation, lifestyle, medical comorbidities, or use of medications [7].

Modelling: The authors assumed that for each country the number of deaths over time (week and year), space and age-sex group follows a Poisson model. That is OK, perhaps the author could try a negative binomial but that is not an issue because that random effects may take into account the overdispersion.

Authors' reply: We thank the reviewer for this comment. We agree with the reviewer that the different

random effects used in the model (weekly, yearly and spatial) are expected to account for overdispersion. In the early stages of this analysis we have tried zero-inflated Poisson and negative binomials and ran the cross validation and the results were identical. As the models with Poisson and the different random effects imposed a lower computational burden, we selected them as our main models in the study.

What was spatial level used? Did the author fit three models per country (National, NUTS2 and NUTS3) and for each model use the forecasted values for 2020, or just one model at NUTS3 level and the NUTS2 and National level statistics were calculated by aggregating the forecasted values for 2020? These two approaches do not lead to the same results for NUTS2 and national level.

Authors' reply: The spatial level used in the analysis was NUTS3 regions and all the forecasted values for 2020 are results of aggregating the estimates at the different levels. We further clarify on page 11, lines 313-316:

NUTS3 regions and weeks were the main spatial and temporal resolution of the analysis. To calculate the different temporal, spatial, sex and age specific aggregations, we combined $p(y_{j,sk6} | \mathcal{D})$ for the different age and sex groups or integrate in time or space accordingly, resulting in the corresponding posteriors.

Temperature: In the proposed model the authors have added temperature as a covariate in a spline-like fashion by using a RW2 model, what was the temperature effect on mortality risk? Is there any? Should the temperature be lagged? It would be nice some comments on this issue. Is winter worst for elder people?

Authors' reply: We thank the reviewer for this comment. We have now added the effect of temperature in the different age and sex groups and countries in the online supplement, Fig.S2-6. We have also added a sentence on the model validation subsection about the temperature effect (page 3, lines 78-80):

In line with the literature [8], the relationship between temperature and mortality was U-shaped, steeper for colder temperatures and stronger for the elderly, Supplementary Fig. S2-6.

Estimating the entire extend of the effect of temperature on all-cause mortality and its different dimensions, for instance spatial effect modification, temporal adaptation, synergistic effects etc. was out of the scope of the study. That also includes the adjustment of the lag-effect, although, as our data is weekly and not daily our approach has a natural lag adjustment for at most 7 days lag. Temperature was rather included to help the model predictions, and the current specification found to have high predictive ability, see cross-validation results in Table S1. We now clarify this in the methods section (page 10, line 277-278):

We are modelling weekly deaths due to data availability, hence we are implicitly considering a 0-7 lag.

Population: The population projections were used essentially as offset in the fitted models, however there are some uncertainty on these projections the further from the last census the bigger the uncertainty. Hence the credible intervals for the mortality risk should be larger if the population uncertainty were propagated in the model. I am not asking to do this, but add as a limitation. There are ways to estimated projected populations in a Bayesian fashion see Raftery et al. (2014) and Gerland et al. (2014)

Authors' reply: The reviewer makes a valid point about the population uncertainty. We now acknowledge this point as a limitation of the study (page 6, lines 154-156):

We used population estimates coming from the national statistical institutes, but we did not account for their uncertainty, which is supposed to increase the further we move from the last census. This is likely to underestimate the width of the 95% CrI of the excess mortality [9].

Excess deaths in other countries/regions/cities: The authors have avoided to discuss excess mortality from countries other than the five chosen ones. I quite understand why, however, it would be interesting in the discussion some values for excess mortality from other countries like the US, Brazil, Mexico, Japan, Korea, France, Sweden, etc. to illustrate how homogeneous (or heterogeneous) are the studied countries, how large or small were the excess deaths in space and time for instance in Madrid compared to Sao Paulo. I am not asking a systematic review, nor the values for all these places, but it would be interesting to see some in order to understand how are the five studied countries compared to other places (in Europe and/or throughout the world).

Authors' reply: We have now added a paragraph in the discussion, discussing our results in the context of other selected countries (page 7, lines 184-195). We are focusing on studies that have reported sub-regional excess mortality, as comparisons at the nationwide level are already available [4, 10]:

Selected studies focusing on regional excess mortality during 2020 in countries not included in the current study have also reported geographical discrepancies [11, 4]. A study focusing in selected Latin America countries found that the subregions of Roriamã, Lima, Puebla and Santa Elena were the most affected in Brazil, Peru, Mexico and Ecuador with the percentage increase in the excess mortality varying from 50 to 160% during 2020 [11]. These values are much higher compared to the maximum observed increase in excess mortality in our study which was 33% in males across all ages in Madrid. Similarly, a lot of geographical variation was observed in the US, with a 35% maximum percentage increase in excess mortality observed in the state of New Jersey during 2020, comparable with the value observed in males in Madrid [4]. Regional studies focusing on the early stages of the pandemic reported geographical discrepancies in Europe and Asia. A study in France reported 63.7% excess mortality in Île-de-France

during March and May 2020 [12]. A study focusing on the three months of the COVID-19 outbreak in China reported 56% (33% – 87%) increase in the excess in three Wuhan districts, with no significant excess in the rest of the country. [13].

References:

Raftery et al. (2014) <https://pubmed.ncbi.nlm.nih.gov/25324591/>

Gerland et al. (2014) <https://www.science.org/doi/10.1126/science.1257469>

References

- [1] Hellenic statistical authority. Data on weekly deaths: Period 1st to last week, 2020, 2021.
- [2] Public Health England. Excess mortality in England: weekly reports, 2021.
- [3] Marília R. Nepomuceno, Ilya Klimkin, Dmitry A. Jdanov, Ainhoa Alustiza Galarza, and Vladimir Shkolnikov. Sensitivity of excess mortality due to the COVID-19 pandemic to the choice of the mortality index, method, reference period, and the time unit of the death series. *medRxiv*, 2021.
- [4] V Kontis, JE Bennett, RM Parks, T Rashid, J Pearson-Stuttard, P Asaria, B Zhou, M Guillot, CD Mathers, YH Khang, M McKee, and M Ezzati. Lessons learned and lessons missed: impact of the coronavirus disease 2019 (covid-19) pandemic on all-cause mortality in 40 industrialised countries prior to mass vaccination. *Wellcome Open Research*, 6(279), 2021.
- [5] Kathryn E Mansfield, Rohini Mathur, John Tazare, Alasdair D Henderson, Amy R Mulick, Helena Carreira, Anthony A Matthews, Patrick Bidulka, Alicia Gayle, Harriet Forbes, et al. Indirect acute effects of the COVID-19 pandemic on physical and mental health in the UK: a population-based study. *The Lancet Digital Health*, 3(4):e217–e230, 2021.
- [6] Paolo Cintia, Luca Pappalardo, Salvatore Rinzivillo, Daniele Fadda, Tobia Boschi, Fosca Giannotti, Francesca Chiaromonte, Pietro Bonato, Francesco Fabbri, Francesco Penone, et al. The relationship between human mobility and viral transmissibility during the COVID-19 epidemics in Italy. *arXiv preprint:2006.03141*, 2020.
- [7] Sunil S Bhopal and Raj Bhopal. Sex differential in COVID-19 mortality varies markedly by age. *The Lancet*, 396(10250):532–533, 2020.

- [8] Antonio Gasparrini, Yuming Guo, Masahiro Hashizume, Eric Lavigne, Antonella Zanobetti, Joel Schwartz, Aurelio Tobias, Shilu Tong, Joacim Rocklöv, Bertil Forsberg, et al. Mortality risk attributable to high and low ambient temperature: a multicountry observational study. *The Lancet*, 386(9991):369–375, 2015.
- [9] Adrian E Raftery, Leontine Alkema, and Patrick Gerland. Bayesian population projections for the United Nations. *Statistical science: a review journal of the Institute of Mathematical Statistics*, 29(1):58, 2014.
- [10] Nazrul Islam, Vladimir M Shkolnikov, Rolando J Acosta, Ilya Klimkin, Ichiro Kawachi, Rafael A Irizarry, Gianfranco Alicandro, Kamlesh Khunti, Tom Yates, Dmitri A Jdanov, et al. Excess deaths associated with COVID-19 pandemic in 2020: age and sex disaggregated time series analysis in 29 high income countries. *BMJ*, 373, 2021.
- [11] Everton EC Lima, Estevão A Vilela, Andrés Peralta, Marília Rocha, Bernardo L Queiroz, Marcos R Gonzaga, Mario Piscocoyá-Díaz, Kevin Martínez-Folgar, Víctor M García-Guerrero, and Flávio HMA Freire. Investigating regional excess mortality during 2020 COVID-19 pandemic in selected Latin American countries. *Genus*, 77(1):1–20, 2021.
- [12] Anne Fouillet, Isabelle Pontais, and Céline Caserio-Schönemann. Excess all-cause mortality during the first wave of the COVID-19 epidemic in France, March to May 2020. *Eurosurveillance*, 25(34):2001485, 2020.
- [13] Jiangmei Liu, Lan Zhang, Yaqiong Yan, Yuchang Zhou, Peng Yin, Jinlei Qi, Lijun Wang, Jingju Pan, Jinling You, Jing Yang, et al. Excess mortality in Wuhan city and other parts of China during the three months of the COVID-19 outbreak: findings from nationwide mortality registries. *BMJ*, 372, 2021.